# Visuo-motor integration, vision perception and attention in mTBI patients. Preliminary findings

**Mariagrazia Benassi[1]\***, **Davide Frattini[1]**, **Sara Garofalo[1]**, **Roberto Bolzani[1]**,
**Tony Pansell[2]**

**1** Department of Psychology, University of Bologna, Bologna, Italy, **2** Department of Clinical Neuroscience, Karolinska Institutet, Stockholm, Sweden

\* mariagrazia.benassi@unibo.it

**Data Availability Statement:** All relevant data are within the paper and its Supporting information files.

**Funding:** The authors received no specific funding for this work.

## Abstract

Patients with mild traumatic brain injuries (mTBI) often report difficulties in motor coordination and visuo-spatial attention. However, the consequences of mTBI on fine motor and visuo-motor coordination are still not well understood. We aimed to evaluate whether mTBI had a concomitant effect on fine motor ability and visuo-motor integration and whether this is related to visual perception and visuo-spatial attention impairments, including patients at different symptoms stage. Eleven mTBI patients (mean age 22.8 years) and ten healthy controls participated in the study. Visuo-motor integration of fine motor abilities and form recognition were measured with the Beery-Buktenica Developmental Test of Visual-Motor Integration test, motion perception was evaluated with motion coherence test, critical flicker fusion was measured with Pocket CFF tester. Visuo-spatial was assessed with the Ruff 2 & 7 Selection Attention Test. mTBI patients showed reduced visuo-motor integration, form recognition, and motor deficits as well as visuo-spatial attention impairment, while motion perception and critical flicker fusion were not impaired. These preliminary findings suggest that the temporary brain insults deriving from mTBI compromise fine motor skills, visuomotor integration, form recognition, and visuo-spatial attention. The impairment in visuo-motor coordination was associated with speed in visuo-attention and correlated with symptoms severity while motor ability was correlated with time since concussion. Given the strong correlation between visuomotor coordination and symptom severity, further investigation with a larger sample seems warranted. Since there appeared to be differences in motor skills with respect to symptom stage, further research is needed to investigate symptom profiles associated with visuomotor coordination and fine motor deficits in mTBI patients.

## Introduction

Mild traumatic brain injury (mTBI) or concussion affects the cognitive abilities of approximately 42 million people worldwide every year [1]. mTBI is defined as a physiological disruption of brain functions, which may occur after a concussion or acceleration/deceleration

**Competing interests:** The authors have declared that no competing interests exist.

movement of the head [2], where despite physical, cognitive, and behavioral symptoms, no evidence of biological injuries appears clearly in medical imaging [3]. Diffuse axonal injuries are considered to be the primary cause of the variety and concurrence of mTBI symptoms [4]. The axonal electrophysiological alteration, in particular, affects the correct transmission of information between sensory organs and the primary sensory cortex, as well as within the cortical areas themselves.

The visual and cognitive disorders generally reported by mTBI patients concern photophobia, visual motion sensitivity, visual and attentional impairments, and behavioral complaints, as being as easily fatigued [5–7]. Concerning visual symptoms, although previous studies demonstrated that despite normal visual acuity, other visual functions are altered in mTBI patients, it is still not clear whether these visual deficits are specific or generalized. Some findings revealed that when first-order low spatial frequency luminance grating stimuli are used, mTBI patients showed normal sensitivity, while deficits appeared for second-order stimuli [8]. In contrast, another study found that mTBI patient's performance (measured as reaction time) was lower for both first and second-order dynamic stimuli [9]. Other findings showed abnormalities for high temporal frequency resolution (i.e. flicker fusion) in the periphery and coherent motion perception deficits [5, 10]. Spiegel et al. [6] analyzed the full contrast sensitivity function for both first- and second-order dynamic and static stimuli. The results showed that mTBI patients have lower sensitivity for orientation-defined and contrast-defined stimuli, but higher spatial frequency sensitivity both for first and second-order stimuli. Based on this evidence, Spiegel et al. [6] proposed that mTBI visual symptoms could be linked to altered temporal processing of visual information, in agreement with previous findings indicating visual symptoms related to flickering [11, 12]. However, the critical flicker frequencies may be related to the severity of light sensitivity symptoms in mTBI patients [13]. Visual symptoms could also be explained as a non-optimal signal-transmission to higher cortical areas, resulting in a decreased ability to discriminate signal to noise. This is in line with those studies indicating elevated thresholds for global motion, as assessed by the random dot kinematogram [5] and impaired adaptation to optic flow [14].

The increase of intracortical excitability could alter not only vision and motor abilities but also higher-order processing such as visuo-motor coordination. Indeed, alteration of excitability in the middle temporal area through transcranial direct current stimulation showed that areas involved in motion perception performance are strongly involved also in visuo-motor coordination [15]. Different studies demonstrated that after concussion visuomotor coordination is usually altered [16, 17]. These studies are based on a double-step task, an experimental paradigm evaluating how the visuomotor system is capable to control the hand position relative to a visual target by using the visual and proprioceptive information to adjust the movements towards the moving target. This ability is seemed to be guided by Posterior Parietal Cortex, PPC, as patients having lesioned these regions are not able to adapt the hand position to sudden changes in target position [18] and, accordingly, healthy participants are not able to adapt the hand response when single-pulse transcranial magnetic stimulation is applied to the PPC [19].

Although vision and visual attentional impairment were observed as typical of mTBI [20], little is known about visuo-motor coordination difficulties. A specific impairment in visuo-constructional ability assessed through VMI-6 [21] was found by Sutton and colleagues [22] in a sample of moderate to severe TBI children with open head injuries. Moreover, mTBI patients often display deficits in the visual processing of complex stimuli [8]. From computational models of the morphology of white matter structures, it was observed that crucial areas, as the superior parietal lobe (SPL) and others in the PPC were more prone to damage in DAI following concussion [3]. This evidence and the role of the V6 complex inside SPL as a visual-input

node to the eye/hand coordination [23] hint at possible deficits in visuo-motor and fine motor ability in mTBI.

Given the results described above, this study aims to investigate whether mTBI affects diffuse motor, visual and cognitive abilities involving fine motor, visuo-motor integration, visuospatial attention, motion perception compromising both lower and higher visual and motor functions. We hypothesize that mTBI disrupts a diffuse visuo-motor network with consequences in reduced visuo-spatial abilities, visual motion perception, motor, and visuo-motor coordination. Moreover, we would investigate whether this altered functioning may reflect the severity of the symptoms. A particular interest of the present study is to analyze the neurocognitive functioning of mTBI patients having symptoms that persist for more than three months after concussion (i.e. having post-concussion syndrome). Indeed, this occurs very often in the mTBI population: up to 64% of mTBI patients still manifest three or more postconcussion symptoms at 3 months [24], while up to 44% report three or more symptoms at 1 year [25]. We expect that having more symptoms is associated with an increased impairment in neurocognitive functions particularly for those patients with persisting symptoms after concussion.

## Materials and methods

### Participants

Participants included 11 mTBI patients (4 females; age range 15–33 years; mean age 22.8 years), and 10 healthy volunteers (2 females; age range 21–26 years; mean age 23 years) used as a control group. Given the small sample size, it was not possible to analyze gender differences in the present study. The patients were referred to St Eric Eye Hospital, Stockholm (Karolinska Institutet, Sweden) by a physiotherapist or a neurologist for evaluation of visual function due to lasting symptoms from a concussion during near work (e.g. blurriness during reading, jumping letters, eye-strain, headache during near work) and visual motion hypersensitivity. The diagnosis of mTBI was initially given by a specialist in brain trauma and neurology in the acute phase at the hospital emergency. The time from the last injury to examination was on median of 148 days (IQR 46–256). All subjects had previous concussions, mostly sport related. The criteria for inclusion in the study for mTBI patients are those specified by the American Congress of Rehabilitation Medicine (1993), and concerned the presence of alterations in the mental state at the time of the accident and focal neurological deficits that could be transient or not. All subjects were symptomatic and had visual complaints that followed the concussion. The patients were given a comprehensive vision examination to assess distance (visual acuity, visual fields) and near visual function (near the point of accommodation, near the point of convergence, eye deviation at distance, and near and grade of stereopsis) as well as the ocular health to exclude the presence of damages to the retina and the visual pathways. All the patients were included in the study because they did not show any damages to the retina and visual pathways. A 15-item questionnaire concerning visual symptoms (Convergence Insufficiency Symptom Survey; CISS) was administered to the patients (see S1 Appendix). Participants had to answer on a Likert scale from 1 (never) to 5 (always). A score ≥ 21 was considered as symptomatic (max score 60) to assess the severity of the visual symptoms. Only one patient scored below 21 (i.e. 19), indicating low visual symptoms. Median CISS was 38 for the mTBI Group (IQR 28–42). All the participants were tested with vision optically fully corrected.

All the participants were recruited voluntarily and were informed of The Code of Ethics of the World Medical Association (Declaration of Helsinki). The privacy rights of subjects have been observed. The study was approved by the Swedish Ethical Review Authority (EPN 2018/ 1768-31/1). Informed consent was obtained from all subjects in written form after having been

informed of the nature and possible consequences of the investigation. Participants aged between 15 and 18, according to Swedish ethical regulations, give their consent if they realize what the research means for him or her. Patients were informed that perceptual tests, especially the motion tests, could produce ailments because of the symptomatology they presented. Everyone was informed that it was possible to take any break and that it was possible to quit the study at any time. Only one mTBI patient discontinued because of too severe symptoms during testing.

## Measures

**Beery-Buktenica Developmental Test of Visual-Motor Integration test (VMI-6).** VMI-6 [21] is a standard paper-pencil test used to assess visuo-motor integration, fine motor, and visual ability. It is composed of three subtests: visuo-motor, visual, and motor coordination subtest. In the visuo-motor subtest, the subject is asked to copy a figure inside a box with no time limit. Twenty-seven figures of increasing difficulty are presented sequentially. Visuo-constructional ability and action planning are requested in this task since the subject is required to copy a figure with no guideline controlling for space distribution or specific figure features, like points of intersection and interlock. The total score is obtained as the number of figures that are correctly copied (maximum score = 27).

The visual subtest consisted of a form recognition task in which the subject was required to match a figure among different possibilities. The number of alternatives increases every block of figures with a minimum of two (first six items), three (from item 5 to item 10), four, six, to eight (last three items). Twenty-seven figures of increasing difficulty are presented subsequently. The total score is obtained as the number of figures correctly recognized (maximum score = 27).

The motor coordination subtest requires the participant to draw a figure by following a guided route defined with narrow borders. Twenty-seven figures of increasing difficulty are presented subsequently. The total score is obtained as the number of figures correctly drawn without crossing the boundaries (maximum score = 27).

**Ruff 2&7 Selection Attention Test (RSAT).** In the RSAT test [26] the subject is required to mark every number two (2) and seven (7) from a matrix of three rows with digits or letters within a time limit of fifteen seconds for every matrix. Twenty matrices are presented and every fifteen seconds the subject is required to shift to the next matrix, for a total duration of five minutes. The subject is supposed to follow the reading line, and no jumps across lines are permitted. The total score is obtained as the number of targets correctly recognized in each specific matrix (digits or letters). Two attentional scores are considered. The digit matrices control attentional speed, implying top-down attentional mechanisms while letter matrices, automatic detection speed, involve bottom-up attentional mechanisms. The accuracy and speed assess the attentional performances of each subject.

**Critical Flicker Fusion (CFF) test.** The CFF threshold was measured by placing by using a flicker tool (Pocket CFF Tester™, Bernell, Mishawaka, USA) at 40 cm from the subjects' eyes. The head was stabilized by using a chinrest. The flicker tool consists of a hand-held box with a circular light area that at 40 cm distance subtends a visual angle of 5.7˚. It works with two different modalities: a progressive change from flickering to steady light (seeing-to-non-seeing; STNS), and from steady to flickering light (non-seeing-to-seeing; NSTS). The CFF threshold was measured with a method of limits (MOL). The reliability of MOL with CFF in assessing the correct temporal processing resolution threshold of the visual system has already been demonstrated [27]. The subject performed three iterations for each mode (STNS and NSTS) in three different positions of the visual field: centrally, centered at 13˚ visual angle to the right,

A

B

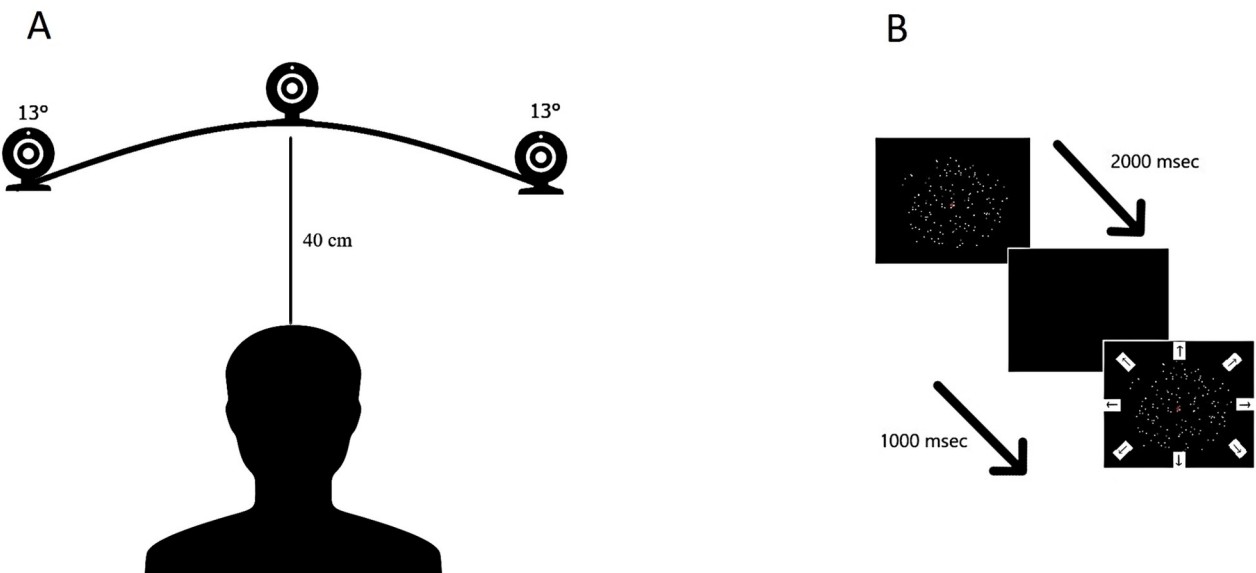

**Fig 1. Experimental setting for the Critical Flicker Fusion test (A) and Motion coherence test (B).** For the Critical Flicker Fusion test, the subject was placed at 40 cm eye-target distance (subtending 5.7˚ visual angle) from the flicker tool with the chin placed on a chinrest. The flicker tool was positioned at three different angles in the visual field as illustrated (centrally, at 13˚ visual angle to the right, and at a 13˚ visual angle to the left). For the Motion coherence test, the subject was placed at 60 cm eye-target distance (subtending 6.2˚ visual angle) from the stimulus.

and centered at a 13˚ visual angle to the left (see Fig 1A). During the iterations in the periphery of the visual field, the subject was required to maintain the gaze on a central fixation target, straight ahead, while releasing the button when detecting the moment of STNS or NSTS. The flickering light was positioned to point toward the subject's eyes. The highest possible frequency of the flicker tool was 60 Hz while the lowest was 35 Hz with a step size of 1 Hz. The change of temporal flickering was 1 Hz/sec. The STNS and NSTS values for each iteration are reported for each subject. The threshold estimation was calculated from the average of all iterations.

**The motion coherence test.** The Motion Coherence Test (see Fig 1B) [28] consists of 150 high luminance dots (luminance 51.0 cd/m2, dot diameter 3 arcmins, dot density 1.25 dots/deg2) that could either move coherently at a constant speed (1.5deg/s) in one of eight directions in the space (four cardinal and four obliques) or in a Brownian manner (noise dots), within a circular frame of 6.2 deg on a black background (0.2 cd/m2) Starting from 100% of coherence, five levels of noise were set such that the signal-to-noise ratio decreased exponentially by two decibels every subsequent level (i.e. by 63.1% in each level).

Therefore, the five levels represent 100%, 63%, 39.8%, 25.1%, and 15.8% of coherence, respectively. Each level is composed of eight trials. Four catch-trials of 100% of coherence were positioned randomly within the task to enhance the subject's attention. Each dot has a limited lifetime of 200 msec. The dot displacement Δx was 4.5 min of arc and frame duration Δt was 50 ms. The monitor refresh rate was 60 fps. The task required the participant to find in what direction, "Where", the coherent dots are moving. Each dot had a limited lifetime of four animation frames (with a duration approximately equal to 200 ms), the stimulus persisted on the screen for 1000 msec, then it disappeared, and all the eight possible directions appeared on the screen. The subject was required to verbalize the direction of the coherent moving dots. The score is obtained as the mean of correct detections in all the coherence levels. For each perceptual test, the subject was trained with a demo version ensuring their comprehension of the

procedure and providing a familiarization with the stimuli. The order of the tests was selected randomized across subjects. All the subjects had a rest between one test and the other.

## Procedure

The tests were randomized across subjects. All the subjects had approximately 5–7 minutes rest between one test and the other to diminish the fatigue effect. The tests were administered in a dim-light and quiet room by an optometrist and a psychologist.

## Data analysis

In order to have more information from data analysis, Frequentist (by using SPSS IBM ver. 25) approach and Bayesian (by using Jasp ver. 0.9) approaches were adopted to test each specific hypothesis. In the Frequentist approach we selected .05 as the threshold of significance (alpha). In the Bayesian approach we adopted a principle of indifference to evaluate the priors in each test (in ANOVA repeated measure analysis we used r scale fixed effect = .5; in Mann-Whitney test a Couchy scale = .707).

To test differences between mTBI and healthy controls on visuo-motor integration, a MANOVA was applied using Group (mTBI vs healthy controls) as the between-subjects factor and the z-scores at VMI-6 subtests as the dependent variable.

To test differences between mTBI and healthy controls on attention, a repeated measure MANOVA was applied using Group (mTBI vs healthy controls) as the between-subjects factor, Condition (automatic detection, controlled search) as a within-subjects factor, and RSAT score as a dependent variable. Mann-Whitney U rank test was applied for RSAT accuracy (because of skewed distribution) using Group (mTBI vs controls) as a between-subjects factor.

To test differences between mTBI and healthy controls on the flicker fusion threshold, a MANOVA for the repeated measure was applied using Group (mTBI vs healthy controls) as the between-subjects factor, eccentricity as the within-subjects factor, and CFF threshold as the dependent variable. To test differences between mTBI and healthy controls on motion coherence accuracy, a Mann-Whitney U test was used because of skewed distribution.

Correlations between the different measures and with symptoms severity evaluated with CISS and time since concussion (TSC days) were analyzed by Spearman's rho correlation analyses.

Data are available as Data in S1 Data.

## Results

With regard to the VMI-6 tasks, the multivariate tests reported a main group effect for all the VMI-6 scales (F(3, 16) = 22.81, p < .001, η2p = .810) (see Fig 2A). Concerning the visuo-motor subtest, a significant difference was observed between groups (F(1, 18) = 20.96, p < .001, η2p = .538); the mTBI group exhibited lower visuo-integration ability (mean ± SEM = 82.5 ± 1.8) than control group (mean ± SEM = 98.6 ± 3.0). This finding was confirmed in Bayesian ANOVA (VMI-6 Visuomotor subtest: Group $BF_{10}$ = 97.609). In the visual subtest the accuracy in discriminate forms was different between groups (F(1, 18) = 5.18, p = .035, η2p = .224), where the mTBI group performed worse (mean ± SEM = 93.1 ± 1.8) than controls (mean ± SEM = 97.6 ± 0.8) (see Fig 2A). Interestingly, the mTBI patients failed only in those items including the highest number of distractors. With Bayesian ANOVA this effect was only partially evident (VMI-6 Visual subtest: Group $BF_{10}$ = 2.148). Finally, in the motor subtest the groups had different performances ($F_{(1, 18)}$ = 19.34, $p < .001$, $\eta^2_p$ = .518), with the mTBI group exhibiting lower motor coordination ability (mean ± SEM = 85.0 ± 2.1) than control group (mean ± SEM = 95.9 ± 1.2). Bayesian ANOVA

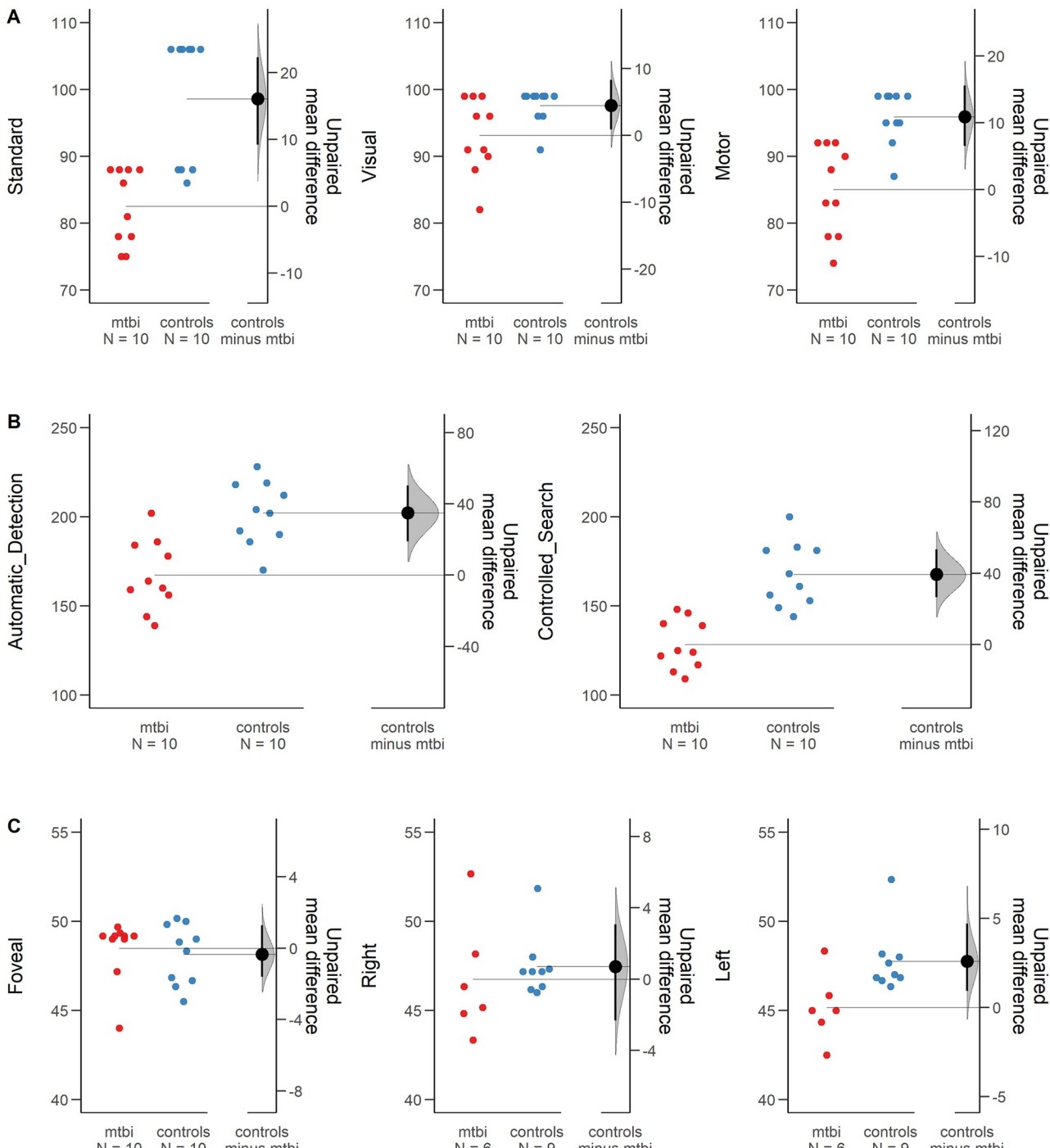

**Fig 2. Visuo-motor, fine motor, form recognition (Visual), and attention skills assessed in mTBI and controls.** (A) VMI-6 test scores indicated difficulties in the mTBI group as compared to the control group for all VMI-6 test speed scales (Error bars indicate standard error mean). (B) The mTBI group exhibited lower performances in both RSAT scales when considering the velocity of response. (C) The mean CFF threshold distribution for mTBI and control group across the visual field indicated difficulties in mTBI as compared to Controls when the stimulus is presented 13˚ Left position as compared to Foveal position.

confirmed a decisive evidence of the difference between group in motor coordination (VMI-6 Motor subtest: Group $BF_{10}$ = 70.360).

With regard to the RSAT scales, the multivariate tests highlighted a main effect of groups ($F_{(2, 17)}$ = 14.45, $p < .001$, $\eta^2_p$ = .630) (see Fig 2B). This effect was highly evident also in Bayesian ANOVA ($BF_{10}$ = 49.642). In the controlled attentional search a significant difference was found between mTBI and healthy controls ($F_{(1, 18)}$ = 29.60, $p < .001$, $\eta^2_p$ = .622) confirming poorer top down attentional ability in mTBI (mean ± SEM = 128.3 ± 4.4) than controls (mean ± SEM = 167.6 ± 5.7). Bayesian ANOVA confirmed extremely high evidence with this respect (Group: $BF_{10}$ = 467.081). Similarly, the automatic detection task was found significant different between the groups ($F_{(1, 18)}$ = 17.12, $p = .001$, $\eta^2_p$ = .448) in which the mTBI group maintained a lower performance (mean ± SEM = 167.2 ± 6.2) in the bottom-up attentional ability, compared to the control group (mean ± SEM = 202.1 ± 5.6) (see Fig 2B). From Bayesian ANOVA the results supported the hypothesis ($BF_{10}$ = 44.137). Lastly, no differences were observed for accuracy, either in the automatic attention task ($U$ = 44.00, $p = .420$, two-tailed) or in the controlled attention task ($U$ = 53.00, $p = .879$, two-tailed). Bayesian Mann-Whitney showed similar findings (Automatic Attention $BF_{10}$ = .406; Controlled Attention $BF_{10}$ = .421).

Concerning the CFF threshold, no significant difference was observed between the two groups ($F_{(1, 13)}$ = 1.46, $p = .248$), whereas a main effect for eccentricity ($F_{(2,12)}$ = 8.82, $p = .004$, $\eta^2_p$ = .595) and the interaction effect between groups and eccentricity was found ($F_{(2, 12)}$ = 7.02, $p = .010$, $\eta^2_p$ = .539). Specifically, the difference observed was significant between the central and peripheral display of CFF ($F_{(1, 13)}$ = 8.17, $p = .013$, $\eta^2_p$ = .386), whilst the within-subjects contrast for the interaction showed a significant difference only between the left and right peripheral position ($F_{(1, 13)}$ = 5.40, $p = .037$, $\eta^2_p$ = .294). As depicted in Fig 2C, the mTBI group exhibited a clear drop in the sensitivity from right periphery (mean ± SEM = 46.75 ± 1.01 Hz) to left periphery (mean ± SEM = 45.1 ± 0.76 Hz), while the control group remained stable from right (mean ± SEM = 47.46 ± 0.82 Hz) to left periphery (mean ± SEM = 47.75 ± 0.62 Hz) (see Fig 2C). Similar results were obtained from Bayesian ANOVA for repeated measure, comparing model containing the effect to equivalent models stripped of the effect, showed differences between groups only when the stimulus was presented in peripheral positions (Group model: $BF_{10}$ = 3.950; Position Model $BF_{10}$ = .634; Group + Position Model: $BF_{10}$ = 2.625; Group + Position + Group*Position Model $BF_{10}$ = 4.230).

The mTBI patients and the control group performed similarly ($U$ = 51.00, $p = .777$) in the motion coherence test. Against the predictions, the mTBI group matched the controls' motion perception performance in terms of accuracy (mTBI mean ± SEM = 0.83 ± 0.28; vs Controls mean ± SEM = 0.81 ± 0.43). Accordingly, the Bayesian Mann Whitney test showed that mTBI and controls had similar performances in the motion coherence test ($BF_{10}$ = .473).

As could be seen in Table 1, Spearman's rho correlation analysis evidenced that VMI Standard score and Motor score were correlated with Ruff2&7 speed attention scores. Symptoms severity evaluated with CISS was negatively correlated with VMI Standard score and time since concussion (days) were positively correlated with VMI Motor score. In detail, a lower visuomotor coordination is associated with an higher symptom severity, an higher time since the concussion is associated with higher motor ability.

## Discussion

The present study demonstrated that mTBI patients present difficulties in visuo-motor integration and fine motor abilities, with concomitant visuo-spatial attention and visual recognition deficits. Coherent motion perception and temporal discrimination abilities seemed preserved when the stimulus is presented foveally. Indeed, for the sample considered in the

**Table 1. Spearman's Rho correlation analysis.**

| | | Mot | VMI Stand | VMI Vis | VMI Mot | R2&7 ADS | R2&7 ADA | R2&7 CSS | R2&7 CSA | CFF Fov | CFF Right | CFF Left | CISS |
|---|---|---|---|---|---|---|---|---|---|---|---|---|---|
| TSC | Rho | 0.215 | 0.245 | -0.370 | **0.642** | -0.115 | 0.150 | 0.115 | 0.273 | 0.263 | 0.029 | 0.116 | -0.085 |
| | p value | 0.526 | 0.494 | 0.292 | **0.045** | 0.751 | 0.679 | 0.751 | 0.445 | 0.462 | 0.957 | 0.827 | 0.815 |
| Mot | Rho | | -0.078 | 0.004 | 0.153 | -0.197 | 0.302 | -0.154 | 0.127 | -0.028 | -0.046 | 0.086 | 0.327 |
| | p value | | 0.744 | 0.985 | 0.520 | 0.405 | 0.196 | 0.517 | 0.593 | 0.905 | 0.871 | 0.760 | 0.356 |
| VMI Stand | Rho | | | 0.381 | **0.514** | **0.582** | -0.141 | **0.663** | 0.136 | 0.230 | -0.009 | 0.275 | **-0.691** |
| | p value | | | 0.097 | **0.021** | **0.007** | 0.552 | **0.001** | 0.568 | 0.330 | 0.976 | 0.321 | **0.039** |
| VMI Vis | Rho | | | | 0.163 | **0.596** | 0.045 | 0.343 | -0.169 | 0.064 | -0.252 | 0.191 | 0.111 |
| | p value | | | | 0.492 | **0.006** | 0.851 | 0.139 | 0.476 | 0.788 | 0.365 | 0.496 | 0.776 |
| VMI Mot | Rho | | | | | 0.400 | 0.180 | **0.643** | 0.087 | -0.087 | 0.238 | **0.735** | -0.128 |
| | p value | | | | | 0.081 | 0.448 | **0.002** | 0.716 | 0.715 | 0.392 | **0.002** | 0.742 |
| R2&7 ADS | Rho | | | | | | 0.054 | **0.806** | -0.086 | -0.012 | 0.038 | 0.368 | -0.350 |
| | p value | | | | | | 0.822 | **<0.001** | 0.717 | 0.961 | 0.894 | 0.177 | 0.356 |
| R2&7 ADA | Rho | | | | | | | 0.185 | 0.171 | **-0.522** | -0.243 | 0.039 | 0.339 |
| | p value | | | | | | | 0.436 | 0.471 | **0.018** | 0.383 | 0.891 | 0.371 |
| R2&7 CSS | Rho | | | | | | | | 0.018 | -0.210 | 0.118 | **0.575** | -0.033 |
| | p value | | | | | | | | 0.940 | 0.374 | 0.674 | **0.025** | 0.932 |
| R2&7 CSA | Rho | | | | | | | | | 0.069 | -0.429 | 0.050 | -0.297 |
| | p value | | | | | | | | | 0.774 | 0.111 | 0.858 | 0.438 |
| CFF Fov | Rho | | | | | | | | | | -0.065 | -0.032 | 0.017 |
| | p value | | | | | | | | | | 0.818 | 0.909 | 0.964 |
| CFF Right | Rho | | | | | | | | | | | **0.573** | 0.800 |
| | p value | | | | | | | | | | | **0.025** | 0.104 |
| CFF Left | Rho | | | | | | | | | | | | 0.500 |
| | p value | | | | | | | | | | | | 0.391 |

TSC, Time since concussion (days); Mot, motion coherence score; VMI Standard, VMI Standard scale; VMI Vis, VMI Visual scale; VMI Mot, VMI Motor scale; R2&7 ADS, R2&7; R2&7 ADA, R2&7; R2&7 CSS, R2&7; R2&7 CSA, R2&7; CFF Fov, Critical Flicker Fusion Test presented foveally; CFF Right, Critical Flicker Fusion Test presented peripherally in the Right visual field; CFF Left, Critical Flicker Fusion Test presented peripherally in the Right visual field; CISS, Convergence Insufficiency Symptom Survey score.

present study, temporal discrimination is anomalous only when presented peripherally 13°, in the left visual field.

Visuo-motor integration relies on correct visuo-constructional functions, action planning, and motor coordination and implies eye-hand coordination [29]. Visuomotor coordination requires a complex control network in which the sensory and motor signals are processed and integrated using feedforward and feedback strategies monitored and guided by the cerebellum and PPC [30]. Even if it has been demonstrated that dynamic corrections of limb movement could occur without the visual feedback about the relative position of the hand [31], the mechanisms involved in VMI-6, as assessed in the present study, strongly implies the coordination between visual and motor system [32]. Studenka & Raiken [32] investigated non-linear dynamics with time series analysis in seventy-five mTBI patients compared to controls facing visuomotor tracking task and found that mTBI patients showed anomalous physiological responses in terms of regularity of the dynamic of motor performance. This regularity is anomalous as compared to controls and seemed to correlate with symptoms severity. Secondly, the authors evidenced gender differences in visuomotor physiological response with mTBI females worse than mTBI males. Abnormal functioning in this network was found also in developmental coordination disorder children who present impaired activation of the

lateral premotor cortex and abnormally increased activation of the prefrontal cortex during the curve tracing task [33]. This could be explained as a consequence of an inefficient functional loop between the parietal cortex and the cerebellum devoted to monitoring forward estimates of limb position and correct ongoing motor commands online could be impaired. In the present study, when performing the visuo-motor integration subtest, all patients were aware of what they saw but still complained about their inability to simultaneously control different aspects of the geometric figure. For example, while drawing one part and simultaneously controlling for interlocks between other elements that form the geometric figure. On the contrary, they did not show any difficulties when drawing 3D figures or figures with intersections that required good visuo-spatial ability but which were less demanding in simultaneous control of information. Patients mostly failed in tasks where visuo-motor constructive abilities were very demanding. The figures were composed of interlocked circles or triangles, requiring the integration of different parts of the visual stimulus into a global shape. mTBI patients failed to appreciate the point of interlock and to parcel the motor command, ending up drawing figures with preserved space distribution but lacking a strategy in completing a correct copy.

The three different VMI-6 subtests revealed three types of impairment. The visual section scores could be interpreted as a selective visual attention deficit since patients showed difficulties in distinguishing figures when they are presented with multiple distractors. Moreover, the VMI motor section revealed a further shortage of fine motor coordination in a task where autonomous visuo-constructional abilities were not requested. The subjects showed lower performance when forced to complete the task in time even though figure boundaries and intersections were already drawn.

These difficulties could rely on general deficits in speed processing in mTBI. Indeed, significant correlations were found between visuo-motor integration and motor performances with speed attention scores. The present finding was partially in contrast with previous evidence indicating independence between visuomotor coordination and attention [18]. However, the former study involved mTBI and ADHD children, showing larger variability both in the motor and attentional abilities.

From RSAT test, mTBI patients were found impaired both for automatic detection and controlled search tasks, when considering the speed parameters. Both VMI and the attentional deficits could hint at executive function anomalies in mTBI patients. Executive functions are needed to optimize visuo-motor coordination and visuo-spatial attention strategies. Previous studies reported executive functions deficit in mTBI [34, 35] and indicated that it is associated with axonal injuries found in the dorsolateral prefrontal cortex [35]. Controlled search demands significant cognitive resources and engages the superior frontoparietal attentional network while automatic detection seems to engage the inferior frontoparietal network [26]. As previously reported, a lack of synchronization between the frontal and parietal attentional neuronal substrates in mTBI might explain both visuomotor and attentional difficulties [7]. A recent study [36] confirmed that already three weeks post-injuries a specific motor learning impairment characterizes mTBI patients.

The mTBI sample considered in the present study did not show any impairment in coherent motion perception and temporal resolution when the stimuli were presented foveally. Patel et al. [5] demonstrated that mTBI patients have higher coherent motion threshold as compared to controls when presenting two matrices of moving dots positioned peripherally. In the present experiment, all motion perception tasks were displayed in the center of the visual field, stimulating the central parafoveal area. Therefore, a possible explanation for the different results could rely on the role of the retinal eccentricity of the stimuli presentation. Accordingly, the significant finding of a change in flicker sensitivity was obtained only in the peripheral position. The connection between temporal resolution processing of the visual

system and motion discrimination impairments has already been demonstrated [37]. Other studies [38, 39] also demonstrated that the segregation of the two visual pathways could be based on the eccentricity of the visual field. We hypothesize that if motion stimuli are displayed in the central retina, with no interference of motion in the surrounding, then the visual processing may cope and avoid the presence of a possible impairment of the dorsal pathway.

A previous study by Schrupp et al. [10] could not demonstrate any difference when comparing the CFF at different retinal eccentricities in mTBI patients but a great variability and general decrease in sensibility in detecting the flicker was noticed. This finding led the authors to conclude that the temporal deficit reflected an impairment in higher visual areas. In the present study, a decreased sensibility in the peripheral visual field was confirmed but also a poor temporal processing resolution in the mTBI group. Although a lateralization effect in flicker discrimination ability has already been demonstrated, with the left inferior parietal lobes (IPL) causally involved in the conscious detection of flicker and the right IPL involved in attention-dependent visual test [40], it is difficult to relate this hemispherical specialization to the preference we found for left visual fields. Previous findings [5] presenting the CFF stimulus foveally in patients with post-concussion syndrome, found higher CFF thresholds associated with higher symptoms severity. In the present study, when the stimulus is presented foveally, no significant relation was found and the Rho coefficient was very low. When the stimulus is presented peripherally the association between CFF threshold and symptom severity was positive and higher, even if not significant. However, the sample presented in Patel et al. [5] study was very different from the one of the present study (e.g. the age ranged from 19 to 72 years and the time since concussion ranged from three months to fifteen years). Further investigations are needed with a larger sample size to confirm these findings, controlling for possible confounding factors (e.g. gender, time since concussion and age). Moreover, although CFF is considered a reliable measure for testing higher visual areas, further studies using cortical activity recording and diffusion tensor imaging while mTBI patients perform perceptive tasks are required to identify the specific areas of disruptions or damage to fiber connections between areas.

This study has limitations mainly related to the small sample size used, which did not allow control for important clinical factors (e.g. symptoms characteristics, pharmacological treatment, gender, and age effects). It is worth noting that the small sample should not have undermined the power, indeed a power analysis was performed expecting a large effect size (G*Power software indicated for independent t-test with alpha = .05; d = 1.5; power = .8 a required sample size 9 subjects for each group), as former findings revealed consistent results of mTBI motor and perceptual impairment.

In conclusion, the relevant deficits found in mTBI patients in visuo-motor integration, motor coordination, and visuo-spatial attention, confirmed the hypothesis about a diffuse impairment beyond a simple perceptual impairment. The pivotal results observed in the present study, could assist the understanding of mTBI visuo-motor and perceptual abilities, indicating that mTBI visuo-motor integration deficits are more related to the attentional difficulties, in terms of speed processing, rather than to the motion perceptual difficulties. Moreover, this study evidenced the short and longterm effects on visuomotor, motor and visual attention in patients with mTBI. Although preliminary, the findings suggest the importance to further investigate how the visuomotor impairments in mTBI could be related to symptoms severity and postconcussion syndrome. Moreover, the results suggested to include a larger sample to investigate possible confounding factors (e.g. the type of symptoms, the time since concussion and the demographic characteristics of the patients, such as gender and age). If confirmed by future studies on larger samples, the results of this study may give indications on

assessment strategies, useful for prognosis and for planning rehabilitation interventions in patients with mTBI.

## Supporting information

**S1 Appendix. Convergence Insufficiency Symptom Survey; CISS.**
(PDF)

**S1 Data.**
(XLS)

## Author Contributions

**Conceptualization:** Mariagrazia Benassi, Davide Frattini, Tony Pansell.

**Data curation:** Mariagrazia Benassi, Davide Frattini, Tony Pansell.

**Formal analysis:** Mariagrazia Benassi, Sara Garofalo, Tony Pansell.

**Investigation:** Tony Pansell.

**Methodology:** Mariagrazia Benassi, Roberto Bolzani, Tony Pansell.

**Project administration:** Mariagrazia Benassi, Tony Pansell.

**Resources:** Tony Pansell.

**Software:** Roberto Bolzani.

**Supervision:** Mariagrazia Benassi, Tony Pansell.

**Validation:** Tony Pansell.

**Visualization:** Tony Pansell.

**Writing – original draft:** Mariagrazia Benassi, Davide Frattini, Tony Pansell.

**Writing – review & editing:** Mariagrazia Benassi, Sara Garofalo, Roberto Bolzani, Tony Pansell.

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
