## [Decision Letter · Decision Letter 0]

9 Dec 2020

PONE-D-20-31588

Visuo-motor integration, vision perception and attention in MTBI patients

PLOS ONE

Dear Dr. Benassi,

Thank you for submitting your manuscript to PLOS ONE. After careful consideration, we feel that it has merit but does not fully meet PLOS ONE’s publication criteria as it currently stands. Therefore, we invite you to submit a revised version of the manuscript that addresses the points raised during the review process.

It seems to me that all the comments are addressable, but however require a significant amount of work. In particular, I would like you to improve the plots. Bar charts tend to minor the difference between the conditions, as the Y axis is probably more expanded than needed. On the contrary, I would recommend using the 95% confidence interval of the mean, instead of SEM, as this is much more informative for the readers (Cumming, 2014).

Cumming, G. (2014). The New Statistics: Why and How. Psychological Science, 25, 7-29.

We look forward to receiving your revised manuscript.

Kind regards,

Robin Baurès, Ph.D.

Academic Editor

PLOS ONE

Journal Requirements:

2.Thank you for including your ethics statement: 

"All the participants were recruited voluntarily and were informed of The Code of Ethicsof the World Medical Association (Declaration of Helsinki). The privacy rights ofsubjects have been observed. The study was approved by the local ethics committee(EPN 2018/1768-31/1)".

3.Please provide additional details regarding participant consent. In the ethics statement in the Methods and online submission information, please ensure that you have specified (1) whether consent was informed and (2) what type you obtained (for instance, written or verbal, and if verbal, how it was documented and witnessed).If the need for consent was waived by the ethics committee, please include this information.

4. Please state in your methods section whether you obtained consent from parents or guardians of the minors (those aged <18) included in the study or whether the research ethics committee or IRB approved the lack of parent or guardian consent.

5.We note that you have indicated that data from this study are available upon request. PLOS only allows data to be available upon request if there are legal or ethical restrictions on sharing data publicly. For more information on unacceptable data access restrictions, please see http://journals.plos.org/plosone/s/data-availability#loc-unacceptable-data-access-restrictions.

6. Please ensure that you refer to Figure 2 in your text as, if accepted, production will need this reference to link the reader to the figure.

Reviewers' comments:

Reviewer's Responses to Questions

**Comments to the Author**

1. Is the manuscript technically sound, and do the data support the conclusions?

Reviewer #1: Yes

Reviewer #2: Partly

2. Has the statistical analysis been performed appropriately and rigorously? 

Reviewer #1: Yes

Reviewer #2: Yes

3. Have the authors made all data underlying the findings in their manuscript fully available?

Reviewer #1: Yes

Reviewer #2: Yes

4. Is the manuscript presented in an intelligible fashion and written in standard English?

Reviewer #1: Yes

Reviewer #2: Yes

5. Review Comments to the Author

Reviewer #1: General comments:

This empirical study examined the visuomotor skill, visual motion perception, and visuospatial attention in young adults both with and without a history of minor traumatic brain injury (mTBI). They observed significant impairments in the visuomotor integration as well as visual motion perception in the mTBI group relative to the control group. The authors discuss their findings in the context of previous findings around brain injury affecting visual and visuomotor areas of the brain. I found literature review and rationale solid, the methods sound, the discussion thorough, and the writing clear. I have a few concerns around clarity of the hypotheses (not stated) and data analysis.

Specific comments:

Intro: Line 45: This would be an appropriate place to spell out your hypothesis, given that this appears to be set up as a hypothesis-driven study and there are adequate previous findings to set up a novel set of hypotheses for the present study. There are two references to hypotheses in the text (data analysis, line 205), but nothing actually spelled out. Indeed, the final concluding paragraph is the first place one sees a sentence describing a hypothesis!

It is not until you get to the methods section that you realize the median time since concussion is many weeks. Thus, this is a study that focusses on the ‘persistent symptom’ situation (which some authors refer to as post-concussion syndrome, but there is a lack of clarity in the field at the moment around the parameters for these categories). Perhaps the authors should have a few sentences in the introduction on this, and make it clear in the abstract that this is a study using a persistent symptom population versus an acute post-injury population. This would be useful to the reader.

Methods/results:

Were there any initial analyses of sex-related differences in your data set. Or, given the sample sizes, was there not enough power for such analyses? Regardless, some mention should be made if an analysis was done or not. This is now standard procedure in biomedical research.

Were there any tests applied to examine dependent variable values as a function of time since concussion, within the concussion group? Given the range (if I’m interpreting the 148 days IQR 46-256 correctly) in time since concussion as well as the range in symptomology, I and other readers would be interested if there is a time effect within the concussion group for your measures.

Discussion

On line 266 there is mention of correlations. This is the first and only reference to correlations between measures. Should a description of this analysis and findings be in the text somewhere?

Lines 282-284 seem oddly placed and redundant, since there is a conclusions section at the very bottom. Also, the phrase eye-motor coordination is confusion, since one is not sure if you’re referring to eye-hand coordination or oculomotor control. But this whole paragraph could be deleted.

Minor:

1.Abstract: “These preliminary findings suggest that the temporary brain insults deriving from mTBI compromise seriously fine motor and visuo-motor integration together with form recognition and visuo-spatial attention.”

Phrasing odd (it sounds like you’re referring to ‘seriously fine motor’ and then there’s no noun). Perhaps just ‘…mTBI compromise fine motor skills, visuomotor integration, form recognition, and visuo-spatial attention’.

2. Abstract:” The unexpected results concerning motion perception could be explained by the position of the stimulus that interested the fovea parafovea whilst in previous studies the stimuli were presented in the peripheral visual field.” Something is missing in this sentence.

3. Intro, line12: “…such as easily fatigued…”, such as being easily fatigued

4. Discussion, line 253: “..that required a good space-control..”. Unclear what this means.

5. Discussion line 280: “… confirmed that already teo weeks post-injuries…” ten weeks post-injury (presumably)

Reviewer #2: This study aims to better understanding the mechanisms that visuo-motor imparments in mild traumatic brain injuries (mTBI) patients. From a behavioural approach, visuo-motor integration of fine motor abilities, motion perception and visuo-spatial attention were measure with several tests. The results coming from 11 mTBI patients were compare to a healthy control group. Dealing with the visuo-motor integration, the form recognition, the motor deficits and the visuo-patial attention, the results are in line with the authors’ main hypotheses and highlight an impairment of these functions in mTBI patients. At the opposite, motion perception was not altered in mTBI patients.

Overall, this study is interesting. The scientific questions raised in their clinical and fundamental context appear relevant. Nevertheless, some major points could significantly improve this work.

1/ The introduction section has a very clinical coloring. Yet the study is conducted with only 11 patients. This is not crippling in itself, but a state-of-the-art approach more focused on the field of visuo-motor control and its underlying mechanisms seems to me more appropriate with the experimental content. Some references of works and models developed by M.Desmurget, Y.Rossetti, or C.Prablanc… appeared missing to me. Moreover, the results of present study could be discussed in the light of these references which are lacking, in order to make the discussion even more attractive.

2/ - My major concerns relate to the presentation of the results, the style of chosen figures, the reported variability, and the nomenclatures displayed. Indeed, the graph bar are very bad representation for a sample of 11 or 10. They do not make it possible to account for variability and distribution. Whisker plot or scatter plot appear much more relevant.

- Graphical visualization of statistical effects would greatly assist the reader.

- Fig 1 and 2 should be grouped together in a same figure with two panels.

3/ The discussion section appeared really descriptive to me. The parallel between the results and the theoretical models of visuomotor functions could be reinforced.

4/ What are the authors' recommendations for perspective of such a work in light with these results?

5/ In conclusion, this work is interesting and can be significantly strengthened. Concerning a major but acceptable limit, namely the sample, the work would be reinforced with a more detailed analysis and representation of the results, and not large averages in graphbars which may mask some fine effects to investigate or at least to comment on. “Preliminary” or “pilot study” could be clearly stated in the title.

6. PLOS authors have the option to publish the peer review history of their article (what does this mean?). If published, this will include your full peer review and any attached files.

Reviewer #1: **Yes: **Lauren E Sergio

Reviewer #2: No

---

## [Author Response · Author response to Decision Letter 0]

16 Mar 2021

Editor’s comments

It seems to me that all the comments are addressable, but however require a significant amount of work. In particular, I would like you to improve the plots. Bar charts tend to minor the difference between the conditions, as the Y axis is probably more expanded than needed. On the contrary, I would recommend using the 95% confidence interval of the mean, instead of SEM, as this is much more informative for the readers (Cumming, 2014).

1. We have modified the manuscript and files according to PlosOne’s style requirements, we have included more information about the Ethic statement, and we present the results following Cumming’s (2014) suggestions.

2.Thank you for including your ethics statement: 

"All the participants were recruited voluntarily and were informed of The Code of Ethics of the World Medical Association (Declaration of Helsinki). The privacy rights of subjects have been observed. The study was approved by the local ethics committee (EPN 2018/1768-31/1)".

2. The text is changed and now includes the name of the ethics committee (see p. 6). 

"All the participants were recruited voluntarily and were informed of The Code of Ethics of the World Medical Association (Declaration of Helsinki). The privacy rights of subjects have been observed. The study was approved by the Swedish Ethical Review Authority (EPN 2018/1768-31/1)".

3.Please provide additional details regarding participant consent. In the ethics statement in the Methods and online submission information, please ensure that you have specified (1) whether consent was informed and (2) what type you obtained (for instance, written or verbal, and if verbal, how it was documented and witnessed).If the need for consent was waived by the ethics committee, please include this information.

 4. Please state in your methods section whether you obtained consent from parents or guardians of the minors (those aged <18) included in the study or whether the research ethics committee or IRB approved the lack of parent or guardian consent.

3 & 4. The participants gave informed consent in written form, and this has now been specified in the methods section. For those participants aged <18, the informed consent was obtained from the minors (see p. 6).

"Informed consent was obtained from all subjects in written form after having been informed of the nature and possible consequences of the investigation. Participants aged between 15 and 18, according to Swedish ethical regulations, give their own consent if they realize what the research means for him or her.”

5.We note that you have indicated that data from this study are available upon request. PLOS only allows data to be available upon request if there are legal or ethical restrictions on sharing data publicly. For more information on unacceptable data access restrictions, please see http://journals.plos.org/plosone/s/data-availability#loc-unacceptable-data-access-restrictions. 

5. We apologize for this oversight and we are happy to provide the original data set, which we will upload with the revised manuscript as Supporting Information files. 

6. Please ensure that you refer to Figure 2 in your text as, if accepted, production will need this reference to link the reader to the figure.

6. We thank the Editor for the indication, we have included in the text the correct reference to Figure that now is numbered as Figure 1B, as suggested by reviewer #2 (see p. 8).

7. We thank the Editor for the indication, we have added captions for Supporting Information files and updated in-text citation.

Response to Reviewers

Reviewer #1:

This empirical study examined the visuomotor skill, visual motion perception, and visuospatial attention in young adults both with and without a history of minor traumatic brain injury (mTBI). They observed significant impairments in the visuomotor integration as well as visual motion perception in the mTBI group relative to the control group. The authors discuss their findings in the context of previous findings around brain injury affecting visual and visuomotor areas of the brain. I found literature review and rationale solid, the methods sound, the discussion thorough, and the writing clear. I have a few concerns around clarity of the hypotheses (not stated) and data analysis.

We thank the reviewer for his/her suggestions, we respond to each comment below.

Specific comments:

Intro: Line 45: This would be an appropriate place to spell out your hypothesis, given that this appears to be set up as a hypothesis-driven study and there are adequate previous findings to set up a novel set of hypotheses for the present study. There are two references to hypotheses in the text (data analysis, line 205), but nothing actually spelled out. Indeed, the final concluding paragraph is the first place one sees a sentence describing a hypothesis!

We specify better in the Introduction the aim and hypothesis of our study (see p. 5 lines 100-113).

It is not until you get to the methods section that you realize the median time since concussion is many weeks. Thus, this is a study that focusses on the ‘persistent symptom’ situation (which some authors refer to as post-concussion syndrome, but there is a lack of clarity in the field at the moment around the parameters for these categories). Perhaps the authors should have a few sentences in the introduction on this, and make it clear in the abstract that this is a study using a persistent symptom population versus an acute post-injury population. This would be useful to the reader.

We thank the reviewer for the comment, we have added in the Introduction and in the abstract a comment on the specific population that we refer to (see p.5 lines 105-110). 

Methods/results:

Were there any initial analyses of sex-related differences in your data set. Or, given the sample sizes, was there not enough power for such analyses? Regardless, some mention should be made if an analysis was done or not. This is now standard procedure in biomedical research.

We confirm that it was not possible to evaluate gender differences because of the small sample size included in the study (only 2 females in the control group). We included a paragraph mentioning this limitation in the method section (p. 5 line 118).

Were there any tests applied to examine dependent variable values as a function of time since concussion, within the concussion group? Given the range (if I’m interpreting the 148 days IQR 46-256 correctly) in time since concussion as well as the range in symptomology, I and other readers would be interested if there is a time effect within the concussion group for your measures.

We thank the reviewer for the comment. We included an analysis based on Spearman’s rho correlation in the results section and a table (Table 1) showing the correlation between the measures used in the present study and time since concussion and symptoms severity. Although the limitation due to the small sample size, the results showed that time since the concussion is related and symptoms severity are correlated with visuomotor and motor ability assessed with VMI-6 (p. 13 line 312 and p.14).

Discussion

On line 266 there is mention of correlations. This is the first and only reference to correlations between measures. Should a description of this analysis and findings be in the text somewhere?

We thank the reviewer for the comment. We included in the results section the correlations based on Spearman’s rho correlation analysis, and the values are visible in Table 1 (p. 13 line 312 and p.14).

Lines 282-284 seem oddly placed and redundant, since there is a conclusions section at the very bottom. Also, the phrase eye-motor coordination is confusion, since one is not sure if you’re referring to eye-hand coordination or oculomotor control. But this whole paragraph could be deleted.

We thank the reviewer for the comment. We agree that the paragraph is redundant, therefore we have deleted it, as suggested.

Minor:

1.Abstract: “These preliminary findings suggest that the temporary brain insults deriving from mTBI compromise seriously fine motor and visuo-motor integration together with form recognition and visuo-spatial attention.”

Phrasing odd (it sounds like you’re referring to ‘seriously fine motor’ and then there’s no noun). Perhaps just ‘…mTBI compromise fine motor skills, visuomotor integration, form recognition, and visuo-spatial attention’.

We thank the reviewer for the suggestion. We corrected the abstract accordingly.

2. Abstract:” The unexpected results concerning motion perception could be explained by the position of the stimulus that interested the fovea parafovea whilst in previous studies the stimuli were presented in the peripheral visual field.” Something is missing in this sentence.

We thank the reviewer for the comment, we deleted the sentence and changed the last paragraph indicating the new results obtained from correlation analysis.

3. Intro, line12: “…such as easily fatigued…”, such as being easily fatigued

We thank the reviewer for the comment we have corrected the sentence (p.3, line 61).

4. Discussion, line 253: “..that required a good space-control..”. Unclear what this means.

We clarify the sentence (p. 15, line 356) as follows: “On the contrary, they did not show any difficulties when drawing 3D figures or figures with intersections that required good visuo-spatial ability but which were less demanding in simultaneous control of information.”

5. Discussion line 280: “… confirmed that already teo weeks post-injuries…” ten weeks post-injury (presumably)

We thank the reviewer for the comment we have corrected the typo (p. 15, line 356).

Reviewer #2

This study aims to better understanding the mechanisms that visuo-motor impairments in mild traumatic brain injuries (mTBI) patients. From a behavioural approach, visuo-motor integration of fine motor abilities, motion perception and visuo-spatial attention were measure with several tests. The results coming from 11 mTBI patients were compare to a healthy control group. Dealing with the visuo-motor integration, the form recognition, the motor deficits and the visuo-spatial attention, the results are in line with the authors’ main hypotheses and highlight an impairment of these functions in mTBI patients. At the opposite, motion perception was not altered in mTBI patients.

Overall, this study is interesting. The scientific questions raised in their clinical and fundamental context appear relevant. Nevertheless, some major points could significantly improve this work.

1/ The introduction section has a very clinical coloring. Yet the study is conducted with only 11 patients. This is not crippling in itself, but a state-of-the-art approach more focused on the field of visuo-motor control and its underlying mechanisms seems to me more appropriate with the experimental content. Some references of works and models developed by M.Desmurget, Y.Rossetti, or C.Prablanc… appeared missing to me. Moreover, the results of present study could be discussed in the light of these references which are lacking, in order to make the discussion even more attractive.

1/We thank the reviewer for the suggestions. We included more information and references on the field of visuo-motor control in the introduction (p. 4 lines 83-90) and discussion (see p. 14-15, lines 335-362).

2/ - My major concerns relate to the presentation of the results, the style of chosen figures, the reported variability, and the nomenclatures displayed. Indeed, the graph bar are very bad representation for a sample of 11 or 10. They do not make it possible to account for variability and distribution. Whisker plot or scatter plot appear much more relevant.

- Graphical visualization of statistical effects would greatly assist the reader.

- Fig 1 and 2 should be grouped together in a same figure with two panels.

2/ We thank the reviewer for the suggestions. We modified the figures as requested.

3/ The discussion section appeared really descriptive to me. The parallel between the results and the theoretical models of visuomotor functions could be reinforced.

3/ We agree with the reviewer comments, we would be more cautious in interpreting the results because of the small sample size, however, we try to improve the discussion section and reinforce the theoretical models of visuomotor functions as suggested (see p. 14-16).

4/ What are the authors' recommendations for perspective of such a work in light with these results?

4/We included final recommendations in the discussion session as follows:

“Although preliminary, the findings suggest the importance to further investigate how the visuomotor impairments in mTBI could be related to symptoms severity and postconcussion syndrome. Moreover, the results suggested to include a larger sample to investigate possible confounding factors (e.g. the type of symptoms, the time since concussion and the demographic characteristics of the patients, such as gender and age). If confirmed by future studies on larger samples, the results of this study may give indications on assessment strategies, useful for prognosis and for planning rehabilitation interventions in patients with mTBI.”

 5/ In conclusion, this work is interesting and can be significantly strengthened. Concerning a major but acceptable limit, namely the sample, the work would be reinforced with a more detailed analysis and representation of the results, and not large averages in graph bars which may mask some fine effects to investigate or at least to comment on. “Preliminary” or “pilot study” could be clearly stated in the title.

5/ We thank the reviewer, we modified the description of the results including whisker plots and distribution plots and we included in the title the statement “Preliminary findings”.

---

## [Decision Letter · Decision Letter 1]

12 Apr 2021

Visuo-motor integration, vision perception and attention in mTBI patients. Preliminary findings.

PONE-D-20-31588R1

Dear Dr. Benassi,

We’re pleased to inform you that your manuscript has been judged scientifically suitable for publication and will be formally accepted for publication once it meets all outstanding technical requirements. Please have a final look at the second reviewer's comment, who recommends a very minor change.

Kind regards,

Robin Baurès, Ph.D.

Academic Editor

PLOS ONE

Additional Editor Comments (optional):

Reviewers' comments:

Reviewer's Responses to Questions

**Comments to the Author**

1. If the authors have adequately addressed your comments raised in a previous round of review and you feel that this manuscript is now acceptable for publication, you may indicate that here to bypass the “Comments to the Author” section, enter your conflict of interest statement in the “Confidential to Editor” section, and submit your "Accept" recommendation.

Reviewer #1: All comments have been addressed

Reviewer #2: All comments have been addressed

2. Is the manuscript technically sound, and do the data support the conclusions?

Reviewer #1: Yes

Reviewer #2: Yes

3. Has the statistical analysis been performed appropriately and rigorously? 

Reviewer #1: Yes

Reviewer #2: Yes

4. Have the authors made all data underlying the findings in their manuscript fully available?

Reviewer #1: Yes

Reviewer #2: Yes

5. Is the manuscript presented in an intelligible fashion and written in standard English?

Reviewer #1: Yes

Reviewer #2: Yes

6. Review Comments to the Author

Reviewer #1: (No Response)

Reviewer #2: The work has been greatly improved. Thanks for working on the figures which present your data so much better for the reader. I really appreciated this version.

May be that further attention is needed to the following point:

Abstract: Line 35 “Visuo-spatial attention was assessed with Ruff2&7”.

It seems abrupt to me to directly enter the short name of the test for unfamiliar readers. Perhaps this sentence should be:

“Visuo-spatial was assessed with the Ruff 2 & 7 Selection Attention Test. “

7. PLOS authors have the option to publish the peer review history of their article (what does this mean?). If published, this will include your full peer review and any attached files.

Reviewer #1: **Yes: **Lauren E Sergio

Reviewer #2: **Yes: **Lilian FAUTRELLE

---

## [Editor Report · Acceptance letter]

16 Apr 2021

PONE-D-20-31588R1 

Visuo-motor integration, vision perception and attention in mTBI patients. Preliminary findings. 

Dear Dr. Benassi:

I'm pleased to inform you that your manuscript has been deemed suitable for publication in PLOS ONE. Congratulations! Your manuscript is now with our production department. 

Kind regards, 

on behalf of

Dr. Robin Baurès 

Academic Editor

PLOS ONE